# Multi-method strategies for provenance determination of coarse-grained igneous rocks: Non-destructive, portable, and quantitative approaches

Bongsu Chang[1], Tae Gun Jo[2], Young Jae Lee[1*]

1 Department of Earth and Environmental Sciences, Korea University, Seoul, Republic of Korea,
2 Research Institute of Buddhist Cultural Heritage, Seoul, Republic of Korea

* youngjlee@korea.ac.kr

## Abstract

Determining the provenance of stones used in cultural heritage artifacts requires interdisciplinary research that integrates archaeology, geology, and analytical science. In this study, we determined the provenance of the source rock for a Bodhisattva stone sculpture from an abandoned temple site in Haman, South Korea. Non-destructive, portable, in-situ multi-analytical methods were used to quantitatively analyze coarse-grained igneous rocks, incorporating macroscopic observations, portable X-ray fluorescence (pXRF), and magnetic susceptibility measurements. Calibration with matrix-matched in-house rock standards and quality checks ensured the reliability of pXRF data for major, minor, and several trace elements, supporting accurate provenance identification. Analysis of spatial variations in chemical composition revealed two distinct geochemical trends across plutonic rock bodies spanning tens of kilometers, providing a key strategy to narrowing down the source area. Our findings present a robust methodology with broad applicability for investigating plutonic rock provenance.

## Introduction

Plutonic rocks have served as common resources for artifact creation since prehistoric times. They were highly valued for remarkable durability, consistent homogeneity, and substantial volume for sufficient supply [1–9]. The geological evolution of these rocks, shaped by the gradual cooling of magma in a subsurface, results in a coarse-crystalline phaneritic texture. This slow cooling process leads to several millimeters in crystal size [10]. This characteristic has significantly impacted the reliability of analytical outcomes in the rock characterization process, particularly in the context of non-destructive analyses, prompting discussions on mitigation strategies [11–14]. Moreover, it is important to understand that spatial variations in chemical and mineralogical compositions can occur within a single plutonic rock mass. These variations are influenced by the differentiation of the emplaced magma and partial heterogeneities in temperature, pressure, and redox conditions [15–18]. When targeting source

**Data availability statement:** All relevant data are within the paper and its Supporting Information files.

**Funding:** This work was supported by the National Research Foundation of Korea (NRF) grant funded by the Korea government (MSIT) (grant no. RS-2024-00345589, recipient YJL).

**Competing interests:** The authors have declared that no competing interests exist.

stones, therefore, it is reasonable to adopt an approach focused on specific outcrops or GPS-based localities at the stage of literature review and subsequent field investigations. This approach is more effective than depending on a generalized strategy based merely on geological regions or petrological classifications.

Rock characterization involves the evaluation of mineral assemblage, texture, physico-chemical properties, and magnetic characteristics. These factors are crucial for accurately determining the provenance of stone material. Despite the accuracy and precision of outcomes, depending solely upon a single technical approach for rock characterization results in fragmented information and limits systematic identification. No single technique can comprehensively capture all the essential characteristics simultaneously. This had led to an increase in provenance studies using multi-technique characterization of archaeological artifacts [19–26]. Specifically, for tangible cultural heritage where sampling and transportation are not feasible, it is essential to employ a multi-analytical, non-destructive, and portable in-situ approach [19,27,28]. Even when a specimen for destructive analysis is available, if it is too small to represent the original object adequately, non-destructive analysis may be preferable because it offers reliable results through broader spatial coverage. In addition, macroscopic examination could be effective and decisive for coarse-grained rocks, where texture is distinctive. The adaptation of statistical approaches to address the limitations of traditional, non-quantitative description-based methodologies is increasing [28–30]. However, a dichotomous approach based on the presence or absence of specific features remains straightforward and effective.

This paper presents a multi-analytical case study on the characterization and provenance of coarse-crystalline igneous stone utilized in the construction of a standing Bodhisattva sculpture in Haman, southeastern Korea (Fig 1). Analyses of the stone

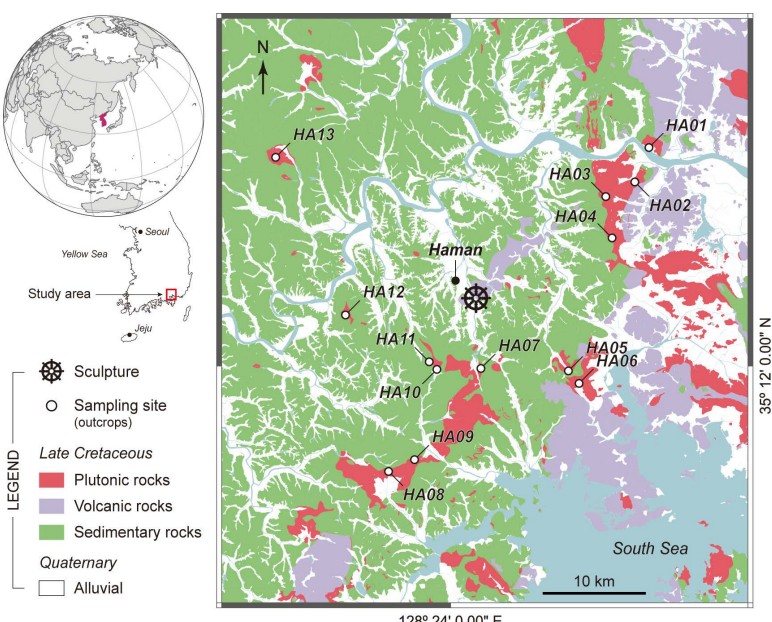

**Fig 1. Geologic map of the study area including Haman and surrounding areas in South Korea.** The map is republished from several digital geological maps [31–41] under a CC BY license, with permission from the Geoscience Data Center of the Korea Institute of Geoscience and Mineral Resources (https://data.kigam.re.kr), original copyrights 1963, 1964, 1969, 1972, 1975, 1983, and 2021.

and its potential source rocks were conducted by integrating X-ray fluorescence (XRF) spectroscopy and magnetic susceptibility (MS) to assess their mineralogical, geochemical, and magnetic attributes comprehensively. Macroscopic observations, with a focus on the detection of enclaves and pinkish alkali feldspar, further informed the evaluation. We aimed to maximize both the efficiency of analysis and the reliability of results by utilizing exclusively non-destructive, portable, and in-situ quantitative analytical methods. This approach reduces ambiguity in the provenance tracking process through quantified evaluation significantly. Additionally, our findings establish a method for defining the range of potential source rocks by detecting systematic trends in chemical composition changes related to spatial distribution in plutonic rock bodies spread over tens of kilometers. This approach is significantly important as it provides an innovative strategy for provenance studies of coarse-grained igneous rocks.

## Materials and methods

### Backgrounds

Since 2010, the Cultural Heritage Administration of the Republic of Korea has been conducting a nationwide research project on abandoned temple sites. Initially, the project focused on evaluating the preservation status of cultural artifacts at risk of damage or loss. In recent, however, it has been reached a turning point, expanding its scope including analysis of raw stone materials and interpretation of their origins. This explanation lays the foundation for further academic research but also deepen our understanding of the historical use and supply of stone materials. It provides objective clues for inferring trade relationships and delivers scientific and rational evidence essential for selecting raw stone materials in the restoration of damaged stone cultural properties, thereby underscoring its significant impact.

The target of this study, the stone Bodhisattva sculpture (Fig 2A), is situated at an abandoned temple site at Daesan-ri, Haman (35° 15' 18.6430" N, 128° 25' 43.3892" E), located in the southeastern Korean Peninsula (Fig 1).

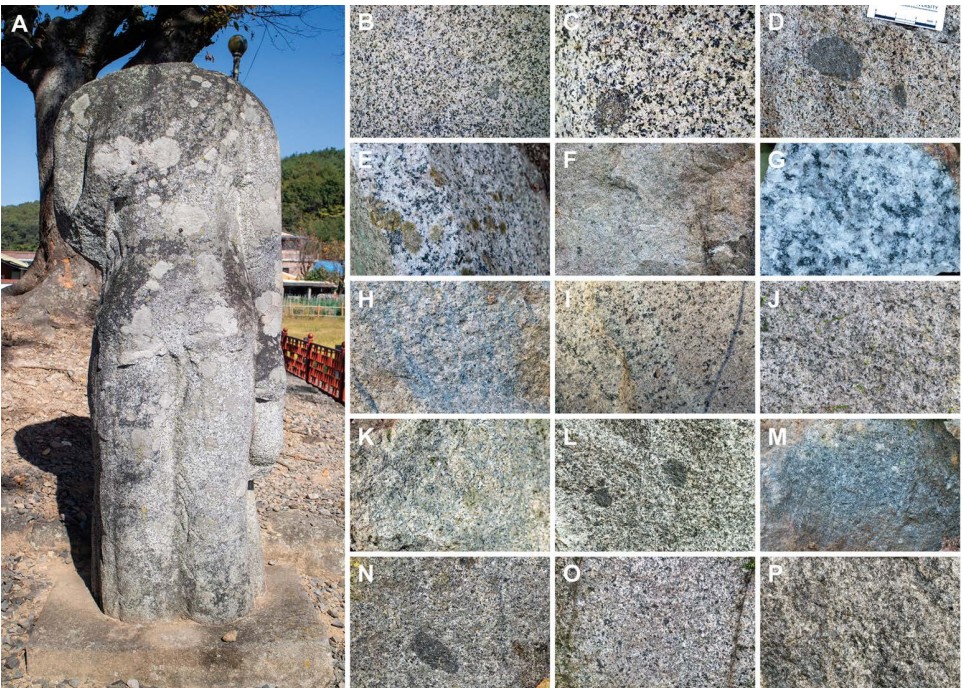

**Fig 2. Photographs of the stone sculpture and outcrop rocks.** (A) Front of the stone sculpture. (B, C) Close-up photographs of the sculpture's surface. (D–P) Photographs of rock surfaces from outcrops HA01 to HA13, arranged sequentially.

Excavations conducted in three phases from 2003 to 2020 unearthed a well remains, roof tiles, utensils, and ceramics, including inlaid celadon and Buncheong ware [42,43]. Based on the art-historical style of these artifacts and the sculpture, as well as their frequency of discovery, the operational period of the temple is estimated to range from the Three Kingdoms period (3rd–7th centuries CE) to the early Joseon Dynasty (~15th century CE), primarily focusing around the mid-7th century to the Goryeo Dynasty (918–1392 CE) [43]. Local Heritage Administration is currently planning to refurbish the area surrounding the sculpture. The sculpture has sustained significant damage, including cracks, flaking, and the loss of its head and right elbow. Therefore, it is urgently necessary to determine the origin of the stone material for restoration. All necessary permits were obtained for the described study from the Cultural Heritage Officer at the Haman County Office, and the fieldwork was conducted in compliance with all relevant regulations.

Geology of the study area is characterized by extensively distributed sedimentary layers formed in terrestrial environments during the Cretaceous period, and also includes andesitic volcanic rocks and acidic plutonic rocks from the late Cretaceous (Fig 1) [44–46]. The sculpture is currently situated in an area where fine-grained andesitic volcanic rock is exposed (Fig 1), contrasting with the coarse-grained igneous rocks that make up the sculpture. In the research area, plutonic rocks are found to the south and east of the sculpture predominantly, indicating that the raw stone material was likely sourced from a site several kilometers distant.

## Problem-solving strategies

The procedure for determining the provenance of the sculpture's source rock, which is the ultimate goal of this study, is structured into three stages (S1 Fig). The first stage involves acquiring scientific data by analyzing the mineralogical, geochemical, and magnetic properties of the sculpture's stone material through macroscopic observations and non-destructive, portable in-situ methods such as magnetic susceptibility measurement and pXRF. Next, a literature review including geological maps enabled us to identify thirteen outcrop locations on plutonic rock bodies within approximately 25 km of the sculpture (Fig 1 and S1 Table). A subsequent field trip examined each potential source rock, facilitating the collection of a non-destructive dataset comparable to the sculpture's analysis data. Finally, by comparing this data with that from the sculpture and assessing the systematic variability in chemical composition of the plutonic rock bodies based on their spatial distribution, we developed practical strategies to pinpoint the rock origins at the outcrop level and to narrow down potential area for sourcing alternative raw stone materials.

## Quantification and statistical analysis

A suite of analyses was conducted using macroscopic observations combined with non-destructive, portable, and field-applicable in-situ instrumental methods designed to produce quantitative data. This approach aimed to minimize practical constraints while maximizing the reliability of the data obtained.

Plutonic rocks consist of coarse-grained mineral phases visible to the naked eye, allowing even non-experts to intuitively assess rock similarity based on the texture of mineral assemblages. Because macroscopic visual observations are inherently qualitative, however, we aimed to minimize misjudgments by applying dichotomous judgment criteria where possible. In igneous rocks dominated by achromatic minerals such as quartz, micas, plagioclase, and amphibole, the presence or absence of alkali feldspars (e.g., orthoclase, microcline) ranging from pale pink to reddish provides a clear dichotomous standard. Similarly, the presence of mafic enclaves, indicative of incomplete mixing of mafic and felsic magmas, also meets this standard. Consequently, in this study, macroscopic observations were systematically documented based on the presence or absence of alkali feldspars and enclaves.

Magnetic susceptibility quantifies the degree to which a material magnetizes in response to an applied magnetic field. The magnitude of magnetic susceptibility is influenced by the type and concentration of magnetic minerals within materials

including magnetite, hematite, ilmenite, titanomagnetite, and goethite. These minerals vary according to rock type and formation environment [47,48]. Due to heterogeneity in mineral distribution, magnetic susceptibility can vary within the same rock type, effectively serving as a unique fingerprint for the outcrop unit rock [27,49,50]. Magnetic susceptibility measurements were conducted using a portable susceptibility meter (KT-10 Plus S/C; Terraplus, Richmond Hill, ON, Canada) with a sensitivity of $1.0 \times 10^{-6}$ SI. Active analysis area is a circle approximately 6.5 cm in diameter, and each measurement takes about 3 sec. Calibration was performed in free air before and after each measurement to ensure accuracy. Considering the inherent heterogeneity of the rock materials, over ten measurements were made at different points for each analysis target.

Whole-rock geochemical composition data were collected using a portable X-ray fluorescence analyzer (Vanta M series; Olympus Evident, Tokyo, Japan). Measurements were performed in GeoChem (2-beam) mode, with each phase lasting 20 sec: phase 1 at 40 kV and phase 2 at 10 kV, totaling 40 sec per analysis point. Each sample object was analyzed a minimum of five times. As previously noted, the target rock's coarse-grained texture requires matrix-matched standards to generate accurate calibration curves for correcting elemental data [11,13,14]. Moreover, pXRF calibration is essential for comparing data across different laboratories and instruments. To meet these requirements, we established a series of calibration curves (S2 Fig) using twenty-five in-house standards, prepared as rock slabs of granite, diorite, and syenite (S3 Fig). The targeted elements included nine oxides ($MgO$, $Al_2O_3$, $SiO_2$, $P_2O_5$, $K_2O$, $CaO$, $TiO_2$, $MnO$, and $FeO$) and eleven metals (V, Cr, Co, Ni, Zn, Rb, Sr, Y, Zr, Nb, and Pb). Elemental reference values for these standards were derived from whole-rock analysis data, which were analyzed using ICP techniques by Activation Laboratories Ltd. (Actlabs) (S2 Table). To guarantee the reliability of the calibration curves, a validation process was conducted using three rock slabs reserved for quality checking, not used during the calibration. Lastly, both absolute and relative errors were evaluated (S3 Table).

## Results

### Macroscopic observations

The stones composing the sculpture vary in color from gray-white to dark blue-gray and exhibit a granular texture. The constituent minerals have a grain size of 3–5 mm, visible to the naked eye (Fig 2A and 2B). The major constituent mineral phases include quartz, plagioclase, biotite, and hornblende, with no evident presence of pinkish alkali feldspar. Characteristically, mafic enclaves measuring approximately 2 × 2 cm or larger are occasionally observed (Fig 2C).

Rocks from the thirteen outcrops displayed varied colors; however, they all shared a mineral grain size of several millimeters, consistent with that of the sculpture's stone (Fig 2B–2P). The evaluation of alkali feldspar and mafic enclaves presence in each outcrop identified the following locations as matching the stone characteristics of the sculptures: HA01, HA04, HA08–11, and HA13 lacked alkali feldspar, and enclaves were identified at HA01, HA09, and HA11. Thus, the rocks at three sites, HA01, HA09, and HA11 satisfied all rock similarity evaluation criteria through macroscopic observations.

### Magnetic susceptibility

The magnetic susceptibility values measured at forty-two spots on the sculpture's surface ranged $18.1–24.9 \times 10^{-3}$ SI, with an average value of $21.2 \times 10^{-3}$ SI, indicative of magnetite series granitoids [51]. Meanwhile, the magnetic susceptibilities for rocks from the thirteen outcrops varied widely, with site averages ranging from $0.15 \times 10^{-3}$ SI to $99.0 \times 10^{-3}$ SI (Fig 3 and S4 Table). Among these, rocks from four outcrops, HA06, HA09–11 aligned with the sculpture's average magnetic susceptibility value, falling within the standard deviation range (Fig 3). Notably, HA09 showed a distribution pattern almost identical to that of the stone sculpture, suggesting that it would be source rock (Fig 3).

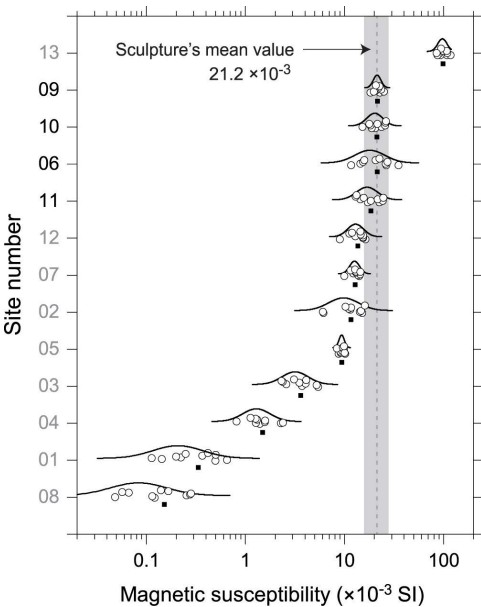

**Fig 3. Distribution of magnetic susceptibility measurements for outcrop rocks by site and their comparison with the sculpture stone's magnetic susceptibility.** Open circle represents individual measurement values, the closed square denotes the average value, and the curved solid line illustrates the log-normal distribution of the data. Vertical dotted line and gray area correspond to the mean value and the range of standard deviation of the sculptural stone, respectively.

## Whole-rock geochemistry

In applying pXRF analysis, we attempted quantitative analysis of twenty elements using the twenty-five matrix-matched standards and prepared three other rock slabs for a quality check (QC) to verify the accuracy of the outcomes. S3 Table presents a summary of pXRF measurements with factory calibration, corrected values using the empirical calibration model for individual elements established in this study. Absolute and relative errors compared against the reference values for the above three QC samples. Chemical composition of major and minor elements, presented in oxide forms, exhibited a relative error of approximately 5% on average, with an absolute error typically around 0.25 wt.%. For trace elements, the relative error averaged about 13%. Considering the limitations of non-destructive and portable in-situ investigation such as matrix effects and surface roughness of stones, and the limited power of X-ray radiation, these error levels are deemed sufficiently accurate for assessing differences and changes in chemical compositions of coarse-grained igneous rocks. These validation results confirm the high reliability of the chemical analyses obtained in this study and highlight the efficiency of pXRF analysis.

Among the quantified twenty elements (S5 Table), eight major and minor ones ($Al_2O_3$, $SiO_2$, $P_2O_5$, $K_2O$, CaO, $TiO_2$, MnO, FeO) were utilized to assess rock similarity, proving effective in distinguishing stones used in sculpture construction (Fig 4). Similarity of chemical compositions between rocks was also evaluated for each element, employing the same methodology as that used for magnetic susceptibility: average compositional value of each candidate rock was deemed consistent only if it fell within the standard deviation range of the sculpture stone. Seven compositions had at least one matching outcrop, and the candidate outcrops that met the criteria showed various combinations for each chemical composition (Fig 4). When examining the relative position of the sculpture stone within the chemical composition spectrum of the thirteen outcrop rocks, it is evident that the stone tends towards lower concentrations in $SiO_2$, $Al_2O_3$, and $K_2O$, while it is positioned in higher ranges for FeO and $TiO_2$ (Fig 4). This pattern is considered to be the results of the differentiation process during magma cooling that formed the plutons.

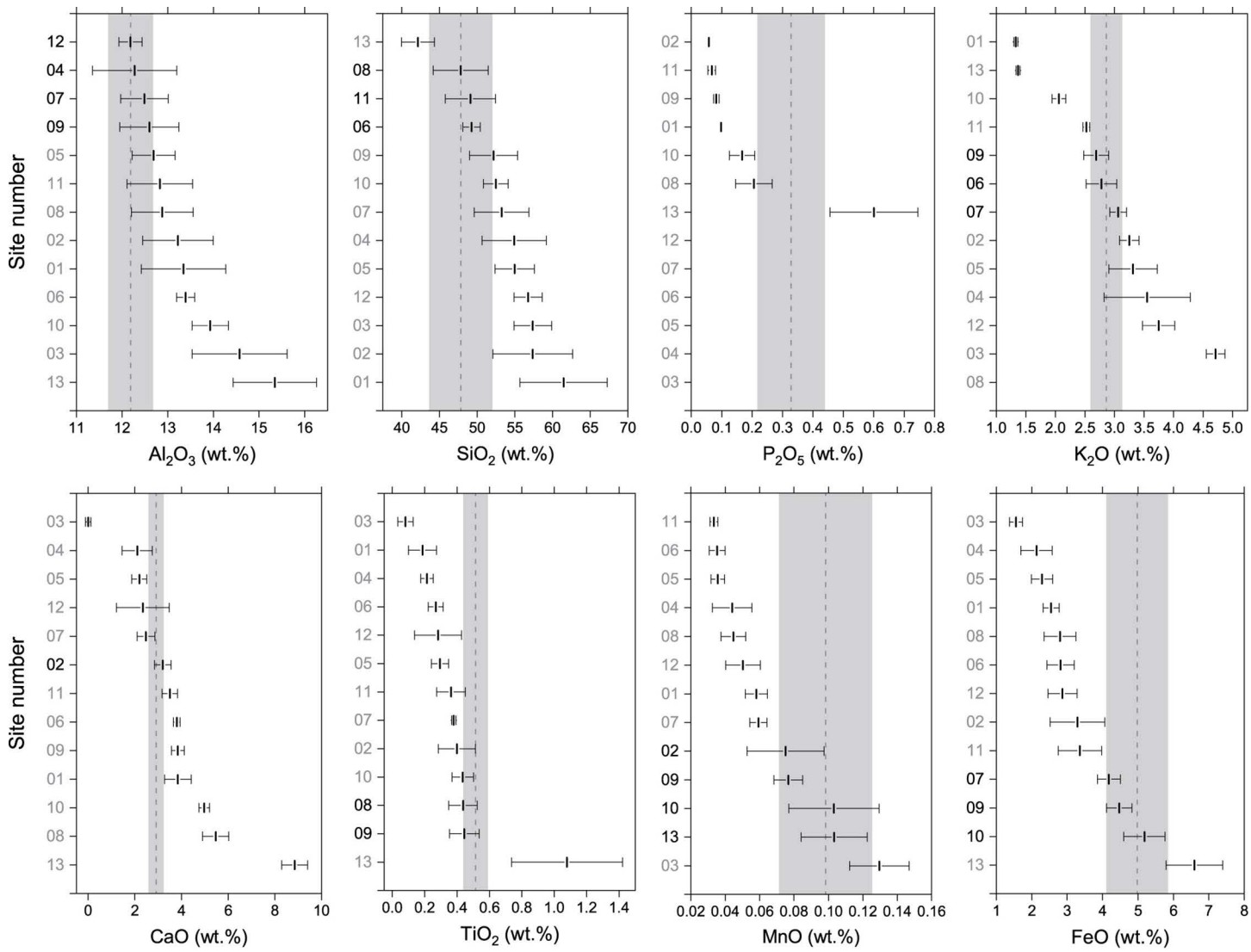

**Fig 4. Ranges of principal geochemical compositions for each outcrop rock and comparison with corresponding elemental contents of a sculptural stone.** Vertical dotted line and gray area corresponding to each chemical composition denote the mean value and the range of standard deviation of the sculptural stone, respectively. Horizontal error bar indicates the standard deviation. Numbers in black on the y-axis represent cases that meet the criteria, while those in gray indicate cases that do not.

## Discussion

### Provenance of sculpture stone at the outcrop level

Analytical methods applied in this study are mutually independent and designed to targeting different characteristics. Therefore, even if a candidate rock meets the criteria of one methodology, passing the standards of other analytical methods cannot be guaranteed. This aspect highlights a unique advantage of multi-analytical investigations in the provenance study. In the previous chapter, the candidate rocks from thirteen exposed outcrops were analyzed for their prominent mineralogical and petrological features, as well as geochemical and magnetic properties. From each analytical approach, outcrop rocks that exhibited similarities to the sculpture stone were identified and picked. By compiling only those that met

the selection criteria for each approach and evaluating them comprehensively, we were able to confidently select the most appropriate source rock.

Fig 5 presents the overall results, showing the status of outcrop rocks that have met the individual criteria for evaluating their mutual similarity. Bar graph represents the cumulative number of elements that matched in the geochemical similarity assessment based on pXRF, while the symbols on the left indicate cases that met the criteria in macroscopic observations and magnetic susceptibility analysis. Detailed information on the symbols and bar graph is provided in the caption. Consequently, it is noted that the outcrop rock at site HA09, which showed the highest degree of similarity, has been assessed as most similar to the stone used in the sculpture construction. Following the HA09 site, which matched in eight criteria, each of HA10 and HA11 matched in four. Given the significant disparity in the number of matches, it is deemed unnecessary to consider ambiguities in the resultant provenance at the outcrop level.

## Effectiveness of coordinating analytical techniques

In this study, rocks were characterized using three independent analytical approaches, enabling us to successfully determine the origin of the source rock at the outcrop level. However, it is necessary to reconsider whether all three types of analytical procedures are essential for successful provenance identification. If the results are consistent, reducing the number of required analyses can enhance efficiency in terms of cost and time.

Here, we classified the possible combinations of the three distinct methods and determined the minimum number of candidate rocks that could be narrowed down in each case (Table 1). Table 1 is presented as a grid showing how each of the three analytical methods (i.e., macroscopic observations, pMS, and pXRF) may be applied individually or in combination. Each cell indicates the number of outcrops selected by that particular combination of methods. The diagonal cells show the number of outcrops meeting the selection criteria when each method is applied alone, whereas the off-diagonal cells show the number selected when two methods are combined. Specifically, macroscopic observations narrowed the selection to three of thirteen outcrops, magnetic susceptibility to four, and pXRF alone to one. In contrast, combining the other two methods (excluding pXRF) did not isolate a single outcrop. Consequently, this study demonstrates that geochemical analysis through pXRF alone can successfully identify a source. These findings emphasize the critical importance of characterizing geochemical properties in determining the provenance of coarse-grained igneous rocks. This

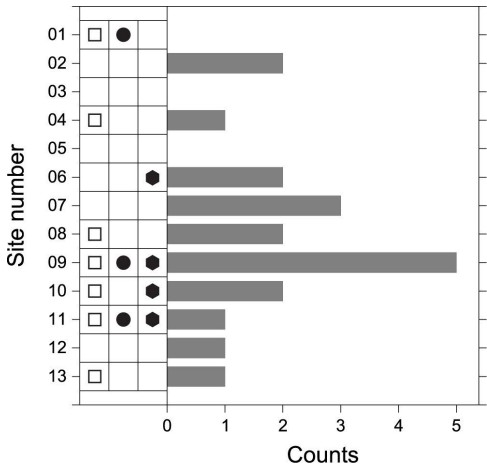

**Fig 5. A plot summarizing the degree of agreement between the analysis results of the stone sculpture and the comparative rocks from thirteen outcrops.** Open square represents rocks lacking alkali feldspars, closed circle denotes rocks containing mafic enclaves, and closed hexagon indicates the case where magnetic susceptibility is matched. A horizontal bar graph displays the cumulative number of matches for each oxide composition, reflecting the geochemical similarity among the eight elements presented in Fig 4.

**Table 1. Possible combinations of analytical methods and the number of outcrops selected for each combination.** The diagonal cells show how many outcrops were passed selection criteria by that single method, while the off-diagonal cells show how many outcrops were selected by the combination of the two analytical methods where the row and column intersect.

| | Macroscopic Observations | pMS | pXRF |
|---|---|---|---|
| Macroscopic observations | 3 | 2 | 1 |
| pMS | 2 | 4 | 1 |
| pXRF | 1 | 1 | 1 |

study also highlights the necessity of employing calibration procedures based on matrix-matched standards to ensure the reliability and accuracy of chemical analyses. Despite the successful results, however, it is important to recognize that these characteristics may be specific to this study. Therefore, it is prudent to employ a comprehensive range of analytical methods that encompass both the physicochemical and magnetic properties of rocks in further research.

## Multidirectional geochemical changes within plutonic rock bodies

Determining rock provenance at the outcrop level offers the advantage of being direct, simple, and tangible due to the specific identification of the target rock. However, when the volume of a selected outcrop is too small or fractured to use the rock mass needed for restoration, or when there are legal, operational, and logistical limitations on access and quarrying, alternatives must be sought. In response to this situation, efforts should be made to find alternatives near the existing outcrop. Nonetheless, if the surface is predominantly covered with vegetation and thick soil layers, which hinders outcrop development, blindly searching for outcrops is impractical. Therefore, a systematic strategy is necessary.

Plutonic rocks form as fluid magma cools slowly over tens of thousands to tens of millions of years [52–54]. During the magma emplacement and cooling processes, mechanisms such as fractional crystallization, assimilation, replenishment, and magma mixing cause differentiation, leading to sequential variations in the bulk chemistry of the magma, typically becoming more silicic [55–57]. These chemical variations are normally associated with spatial changes stemming from the transformation of fluidic melt within the magma chamber into rigid crystals. Consequently, the geochemical characteristics of exposed plutonic rock bodies may exhibit directional trends in their spatial distribution, regardless of mineral assemblages and textural characteristics. In the context of determining the provenance of plutonic rocks, the importance lies in providing a scientific strategy that systematically links geochemical characteristics with spatial changes to delineate the area where rocks with specific geochemical signatures are located. Moreover, when multiple geochemical factors with different azimuthal tendencies exist, the spatial extent that can be delimited becomes narrower, significantly enhancing provenance determination.

Fig 6 shows the distribution of plutonic rock bodies spread over tens of kilometers alongside the locations of thirteen outcrops, demonstrating the systematic variations in specific chemical compositions across the geographical area. For clarity, the HA13 site is excluded from this discussion because it exhibits characteristics that diverge from the observed trends in magnetic susceptibility and geochemical properties at other sites, suggesting a different magma source. According to the relationship between Sr/Y and Y content (Fig 6B), a distinct trend is observed along the SSE–NNW direction. Moving from SSE to NNW, the chemical composition of the rock gradually shifts from the andesite–dacite–rhyolite (ADR) region, typically associated with island arc environments, to the adakite region, formed by the partial melting of deformed basalt that subducted under a volcanic arc. By segmenting the continuous trend in the Sr/Y–Y space with two boundaries (two dotted linear lines in Fig 6A and 6B) across thirteen outcrops into three groups, the sculpture stone is found in the group with the most pronounced ARD characteristics. This classification suggests that the origin of the raw stone material used in sculpture production is confined to the southernmost part of the three geographically segmented areas (Fig 6A). However, a biplot using five major elements has captured a trend of change from WNW to ESE (Fig 6C). Moving in the WNW direction, the combined content of FeO and $TiO_2$ increases, while the sum of $SiO_2$,

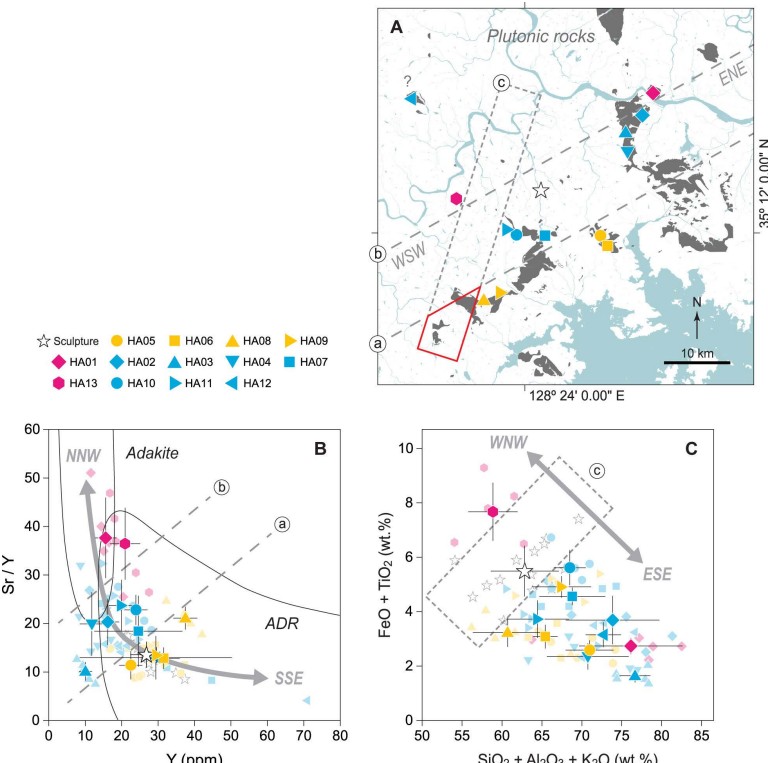

**Fig 6. Constraining the provenance of source rock by analyzing spatial variations in the chemical composition of plutonic bodies.** (A) Spatial distribution of sampling outcrops grouped by its chemical composition and estimation of areas with high chemical affinity to the sculpture's stone based on chemo-geospatial constraints (a red tetragon). (B) Plot of Sr/Y vs. Y showing fields for adakite and island-arc andesite-dacite-rhyolite (ADR) lavas. Boundaries shown as solid curves are those proposed by [58,59]. Gray dotted lines indicate group boundaries based on rock chemical composition. (C) Plot of $SiO_2+Al_2O_3+K_2O$ vs. $FeO+TiO_2$ showing changes in chemical composition depending on the sampling outcrops. A gray dotted rectangle represents the geochemical characteristics of the sculpture's stone (open star) relative to surrounding outcrops. Gray lines with arrows at each end exhibit the azimuth direction on the map. Error bar corresponds to the standard deviation from the mean of the relevant variables. The map is republished from several digital geological maps [31–41] under a CC BY license, with permission from the Geoscience Data Center of the Korea Institute of Geoscience and Mineral Resources (https://data.kigam.re.kr), original copyrights 1963, 1964, 1969, 1972, 1975, 1983, and 2021.

$Al_2O_3$, and $K_2O$ decreases, with the reverse compositional trend observed in the ESE direction. Spatially, the sculpture stone is positioned in a gap between HA13 site and the other twelve sites (Fig 6C). Correspondingly, the geographic area takes on an elongated rectangular shape in the NNE–SSW direction (Fig 6A). As a result, two distinct combinations of chemical compositions provided dual spatial constraints, allowing us to delineate a geographic area that meets both criteria (Fig 6A). This process successfully narrowed the expected source region for the raw stone material for sculpture restoration from tens of kilometers to within a few kilometers. This strategy significantly enhances efficiency by reducing the investigation area, focusing limited resources (i.e., manpower, time, and analytical instrument) on a more targeted area to achieve the final goal. Although the HA09 site, previously identified as the most similar outcrop, is not located within the constrained area, all these areas belong to the same continuous plutonic body as shown on geological map (Figs 1 and 6A) and are in close proximity to each other. Therefore, the strategic value of this approach in narrowing the target area remains undiminished.

## Conclusion

We aimed to determine the provenance of coarse-grained igneous rock used in this sculpture by analyzing their mineralogical, geochemical, and magnetic properties. By not relying on a single property, we reduced uncertainty in provenance

tracing and enhanced the reliability of the results. From an instrumental analysis perspective, we adopted non-destructive, portable in-situ methodologies to minimize constraints, alongside quantitative methods to clarify result interpretations. In this study, pXRF was found to be the most effective technique for determining provenance of coarse-grained igneous rock. The use of matrix-matched in-house standards and subsequent quality checks to validate accuracy ensured the reliability of the pXRF data. In addition, the quantified pXRF data aided in identifying systematic changes in chemical composition related to the spatial distribution of plutonic rock bodies. The multi-directional systematic changes in chemical composition, influenced by tectonic factors and igneous differentiation, played a crucial role in narrowing the target area for tracing rock origins. Consequently, the source rock used in the sculpture appears to have been quarried from a site approximately 20 km away (in a straight line) from its current location. Considering the rugged terrain, the actual transportation route is estimated to have spanned about 35 km, following valleys, inland waterways, and the southern sea. Our methodology introduces a novel strategic approach for provenance studies targeting extensive plutonic rock bodies, highlighting the importance of this research.

## Supporting information

**S1 Fig.  Flowchart showing the sequence and structure of the analytical methods used in this study.**
(TIF)

**S2 Fig.  Calibration curves prepared for the quantitative portable XRF analysis.** The plots represent the correlation between factory-calibrated pXRF data obtained from in-house reference plutonic rock samples (x-axis) and ICP-MS analysis results for the identical samples (y-axis). Red lines indicate linear regression fits. Horizontal error bars represent the standard deviation of five independent measurements. Displayed equations pertain to calibration models for specific elements, with the correlation coefficient ($R^2$) enclosed in parentheses. Units for chemical concentrations are denoted as weight percent (wt.%) for oxides and parts per million (ppm) for metals.
(TIF)

**S3 Fig.  Photographs of selected in-house plutonic rock standards used for matrix-matching calibration.**
(TIF)

**S1 Table.  Geographic location information (GPS coordinates) of thirteen outcrops.**
(XLSX)

**S2 Table.  Reference geochemistry of in-house rock slabs standards.**
(XLSX)

**S3 Table.  Quality-check data for portable XRF analysis.**
(XLSX)

**S4 Table.  Magnetic susceptibility data.**
(XLSX)

**S5 Table.  Chemical composition data.**
(XLSX)

## Acknowledgments

The authors thank Dr. Jieun Seo (Korea University) for participating in the field surveys and her support. The authors appreciate helpful comments and suggestions from Prof. Seon-Gyu Choi (Korea University). The authors gratefully acknowledge instrumental support from Hyun-Ki Lee and Byung-Mok Kim (Youngin AT Corp.). We gratefully acknowledge the insightful comments and constructive suggestions provided by two reviewers. In particular, we thank Robert H. Tykot

for his thorough evaluation, as well as the anonymous reviewer for their valuable feedback, which have greatly contributed to enhancing the quality of this manuscript.

## Author contributions

**Conceptualization:** Bongsu Chang, Young Jae Lee.

**Data curation:** Bongsu Chang.

**Formal analysis:** Bongsu Chang.

**Funding acquisition:** Young Jae Lee.

**Investigation:** Bongsu Chang, Young Jae Lee.

**Methodology:** Bongsu Chang.

**Project administration:** Young Jae Lee.

**Resources:** Tae Gun Jo.

**Visualization:** Bongsu Chang.

**Writing – original draft:** Bongsu Chang.

**Writing – review & editing:** Bongsu Chang, Tae Gun Jo, Young Jae Lee.

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
