## [Decision Letter · Decision Letter 0]

18 Feb 2025

PONE-D-25-03353Multi-method strategies for provenance determination of coarse-grained igneous rocks: Non-destructive, portable, and quantitative approachesPLOS ONE

Dear Dr. Lee,

Thank you for submitting your manuscript to PLOS ONE. After careful consideration, we feel that it has merit but does not fully meet PLOS ONE’s publication criteria as it currently stands. Therefore, we invite you to submit a revised version of the manuscript that addresses the points raised during the review process.

As noted in the reviews appended below, there are some promising results in your paper, but also minor issues that need to be addressed before the paper can go forward. I invite you to resubmit your manuscript after addressing all reviewer comments.

We look forward to receiving your revised manuscript.

Kind regards,

Carlos P. Odriozola, Ph.D

Academic Editor

PLOS ONE

Journal Requirements:

2. In your manuscript, please provide additional information regarding the specimens used in your study. Ensure that you have reported human remain specimen numbers and complete repository information, including museum name and geographic location. 

For more information on PLOS ONE's requirements for paleontology and archeology research, see https://journals.plos.org/plosone/s/submission-guidelines#loc-paleontology-and-archaeology-research.

3. Thank you for stating the following financial disclosure: This work was supported by the National Research Foundation of Korea (NRF) grant funded by the Korea government (MSIT) (grant no. RS-2024-00345589, recipient YJL).

Reviewers' comments:

Reviewer's Responses to Questions

**Comments to the Author**

1. Is the manuscript technically sound, and do the data support the conclusions?

Reviewer #1: Yes

Reviewer #2: Yes

2. Has the statistical analysis been performed appropriately and rigorously? 

Reviewer #1: Yes

Reviewer #2: Yes

3. Have the authors made all data underlying the findings in their manuscript fully available?

Reviewer #1: Yes

Reviewer #2: Yes

4. Is the manuscript presented in an intelligible fashion and written in standard English?

Reviewer #1: Yes

Reviewer #2: Yes

5. Review Comments to the Author

Reviewer #1: This is a fine study on this one stone sculpture, using an appropriate multi-method approach.

You should make clear the sequence of analytical methods used, and if there was any selection in reducing the number of tests.

It should be made clear that pXRF calibration is needed for potential comparison of data between labs and different instruments. In your case, the analyses of multiple spots supported the reliability of your results and interpretation, while accuracy of your elemental data is good for your report for future studies.

Where you discuss Figure 5 (lines 197 onward), need to make clear what the different symbols (open squares, filled circles, filled hexagons) mean. For the pXRF elemental analysis, indicate which elements are represented in the bar graph.

In Figure 6A, where is the sculpture (asterisk)? It should be as clear as you have for 6B and 6C. Or not in the range of this map?

In Conclusion, indicate the distance from where the sculpture is to its geological source.

Are there any other sculptures at this site or elsewhere in the Haman area that might be tested in the future? This would be very informative for any patterns in the use of particular stone for these and other sculptures. Possible future research?

Minor corrections:

line 23: delete "the" and insert "were" after "They"

line 25: insert a word before "subsurface". "a" ?

line 69: Add "the" before "target", and before "stone Bodhisattva"

line 93: add "a" before "geological"

lines 166-167: remove the space between the numbers and % (5%, 13%

line 192: insert "the" before "provenance"

line 202: insert "the" before "HA09 site"

line 216: make clearer what is in Table 1, here in the text and in the caption.

line 232: insert "a" before "selected"

line 282: insert "this" before "sculpture"

Reviewer #2: The research work seems novel to me, as it helps to carry out non-destructive research studies on archaeological materials composed of plutonic rocks with visible crystal sizes. The authors have been able to tackle the problem, using several different methods and understanding the genesis of these plutonic rocks. The text is very well articulated and in very clear language.

Below I point out some formal issues regarding the text:

1. Figure 1 should have a world map to later locate the map of Korea. To avoid confusion, the blue color of the map should be written: Korean Sea.

2. In Figure 6A, the location of the star that indicates the sculpture is not clearly visible.

3. A figure with more detailed photographs of the texture and size of crystals is missing (as far as possible), comparing the rock of the sculpture with the source rock.

4. Has the quarry or signs of quarrying been found? I imagine that this point in question is unlikely. Unless this rock was chosen to build other buildings.

5. Are the sedimentary layers widely distributed around the sculpture formed in terrestrial environments during the Cretaceous period? Are they not typical of marine environments?

6. Is the plutonic rock a tonalite or granodiorite? The mineralogy suggests that it is an intermediate facies.

7. On line 179 it says: “This pattern is considered to be the results of the differentiation process during magma cooling that formed the plutons”. This is true, coinciding with the observation made previously on the presence of enclaves, the abundance of biotite and hornblende, and the scarcity of alkaline feldspar. In principle, more mafic minerals accumulate in the magmatic differentiation process. The existence of biotite and hornblende (rich in Fe and Ti), are a strong argument to explain the high values of FeO and TiO2 existing in the rock corresponding to the sculpture. In parallel, the low presence of alkaline feldspar in this same rock is responsible for the low content of K2O and SiO2. It would have been very good to be able to check the reaction textures in the different enclaves, an issue that is not the subject of this research, since it is only possible through destructive petrographic methods.

6. PLOS authors have the option to publish the peer review history of their article (what does this mean? ). If published, this will include your full peer review and any attached files.

**Do you want your identity to be public for this peer review?** For information about this choice, including consent withdrawal, please see our Privacy Policy .

Reviewer #1: **Yes: ** Robert H. Tykot

Reviewer #2: No

---

## [Author Response · Author response to Decision Letter 1]

26 Mar 2025

Yes, I have included the relevant information in the Response to Reviewers file.

---

## [Editor Report · Decision Letter 1]

21 Apr 2025

Multi-method strategies for provenance determination of coarse-grained igneous rocks: Non-destructive, portable, and quantitative approaches

PONE-D-25-03353R1

Dear Dr. Lee,

We’re pleased to inform you that your manuscript has been judged scientifically suitable for publication and will be formally accepted for publication once it meets all outstanding technical requirements.

Kind regards,

Carlos P. Odriozola, Ph.D

Academic Editor

PLOS ONE
---

## [Editor Report · Acceptance letter]

PONE-D-25-03353R1

PLOS ONE

Dear Dr. Lee,

I'm pleased to inform you that your manuscript has been deemed suitable for publication in PLOS ONE. Congratulations! Your manuscript is now being handed over to our production team.

Kind regards,

on behalf of

Dr. Carlos P. Odriozola

Academic Editor

PLOS ONE